# Prevalence and risk factors of *Salmonella* in commercial poultry farms in Nigeria

Abdurrahman Hassan Jibril[1,2], Iruka N. Okeke[3], Anders Dalsgaard[1], Egle Kudirkiene[1], Olabisi Comfort Akinlabi[3], Muhammad Bashir Bello[4,5], John Elmerdahl Olsen [1] *

1 Department of Veterinary and Animal Sciences, Faculty of Health and Medical Sciences, University of Copenhagen, Copenhagen, Denmark, 2 Department of Veterinary Public Health and Preventive Medicine, Faculty of Veterinary Medicine, Usmanu Danfodiyo University Sokoto, Sokoto, Nigeria, 3 Department of Pharmaceutical Microbiology, Faculty of Pharmacy, University of Ibadan, Ibadan, Nigeria, 4 Department of Veterinary Microbiology, Faculty of Veterinary Medicine, Usmanu Danfodiyo University Sokoto, Sokoto, Nigeria, 5 Centre for Advanced Medical Research and Training, Usmanu Danfodiyo University Sokoto, Sokoto, Nigeria

* jeo@sund.ku.dk

**Data Availability Statement:** The draft genome sequences are available at the European Nucleotide Archive under study accession number PRJEB37477 (secondary accession ERP120792)

## Abstract

*Salmonella* is an important human pathogen and poultry products constitute an important source of human infections. This study investigated prevalence; identified serotypes based on whole genome sequence, described spatial distribution of *Salmonella* serotypes and predicted risk factors that could influence the prevalence of *Salmonella* infection in commercial poultry farms in Nigeria. A cross sectional approach was employed to collect 558 pooled shoe socks and dust samples from 165 commercial poultry farms in North West Nigeria. On-farm visitation questionnaires were administered to obtain information on farm management practices in order to assess risk factors for *Salmonella* prevalence. *Salmonella* was identified by culture, biotyping, serology and polymerase chain reaction (PCR). PCR confirmed isolates were paired-end Illumina- sequenced. Following *de novo* genome assembly, draft genomes were used to obtain serotypes by SeqSero2 and SISTR pipeline and sequence types by SISTR and Enterobase. Risk factor analysis was performed using the logit model. A farm prevalence of 47.9% ($CI_{95}$ [40.3–55.5]) for *Salmonella* was observed, with a sample level prevalence of 15.9% ($CI_{95}$ [12.9–18.9]). Twenty-three different serotypes were identified, with *S*. Kentucky and *S*. Isangi as the most prevalent (32.9% and 11%). Serotypes showed some geographic variation. *Salmonella* detection was strongly associated with disposal of poultry waste and with presence of other livestock on the farm. *Salmonella* was commonly detected on commercial poultry farms in North West Nigeria and *S*. Kentucky was found to be ubiquitous in the farms.

## Introduction

Non-typhoidal *Salmonella* is one of the most common causes of food-borne diseases worldwide. It has been estimated to cause 93.8 million human infections and 155,000 deaths

and accession number for each genome is indicated in S2 File.

**Funding:** This work was part-supported by an African Research Leader Award to INO from the UK Medical Research Council (MRC) and the UK Department for International Development (DFID) under the MRC/DFID Concordat agreement that is also part of the EDCTP2 programme supported by the European Union.

**Competing interests:** Authors declared there is no conflicting interest that could influence the publication of this study.

annually [1, 2]. Contaminated poultry products, especially undercooked meat and raw eggs are important sources of human salmonellosis [3, 4].

Serotyping is the first step to characterize *Salmonella*, because serovars often inform on possible pathogenic potential, host range and disease sequelae [5–7]. Serotyping therefore form the basis of national and international surveillance networks for *Salmonella* [8, 9]. Until recently, traditional serology based on reactions of rabbit antisera to the lipopolysaccharide and flagellar antigens and the surface antigen Vi was used to divide *Salmonella* into more than 2,600 serovars by the White Kauffmann Le Minor Scheme (WKL) [10]. However, whole genome sequencing (WGS) has now emerged as an alternative, rapid and more discriminatory method [7, 8]. By this method, prediction of serotypes can be done using freely available *in silico* pipelines, such as SeqSero, which utilizes surface antigen-encoding genes for predicting serotypes, and *Salmonella In Silico* Typing Resources (SISTR), which infers serovars from core genome MLST (cgMLST) and surface antigens [9, 11, 12]. Several studies have now used WGS in *Salmonella* surveillance and outbreak investigation [2, 13–15], and 91.9% concordance has been found between reported serovars by WKL scheme and predicted serovars using *in silico* resource [16], and 94.8% and 88.2% similarity was reported for SISTR and SeqSero, respectively [17].

Agricultural sector remains the largest contributor to the Nigerian economy, accounting for over 38% of the non-oil foreign exchange earnings, and employing about 70% of the active labour force of the population. The poultry sub-sector is the most commercialized of all the sub-sectors of the Nigeria's agriculture [18] and has transformed the lives of the less privileged segment of the society with just a little investment and at low cost of technology. Annual production average 454 billion tonnes of meat and 3.8 million eggs, with a standing population of 180 million birds [19]. Poultry meat and eggs are the major sources of animal protein in Nigeria, as in many developing countries, because of their affordability and acceptability [20, 21]. Unfortunately, the sustainable growth of this important agricultural subsector is seriously threatened by several infectious diseases including, those caused by *Salmonella* species. So far, there are only a few published reports of circulating strains of *Salmonella* in poultry production in Nigeria [21–24], and very little has been done to understand the risk factors for the different types of *Salmonella*. The aim of the present study was to determine *Salmonella* prevalence, serotype distribution by WGS and risk factors for *Salmonella* obtained in commercial poultry farms in Nigeria.

## Materials and methods

### Ethical approval

Ethical approval for sampling and questionnaire investigation to obtain farm data was obtained from Sokoto State Ministry of Animal Health and Fisheries Developments, Kebbi State Ministry of Animal Health and Husbandry, and Zamfara State Directorate of Animal Health and Livestock Development with approval reference numbers MAH&FD/VET/166/11, MAHF/VET/VOL1, and DAHLD/SUB/VET/VOL.1 respectively.

### Study area

The study was conducted in north-western Nigeria. The region occupies a total land mass of 226,662 km$^2$, representing 24.5% of Nigeria's total land mass. There is an estimated human population of 48,942,307 (25.3% of Nigeria's total population) majority of whom are involved in farming activities [25, 26]. The region also has an estimated exotic and backyard poultry population of 18,770,610 and 10,064,763 which respectively represent 16.2% and 46.2% of the

total chicken populations in these categories in the country [18]. Sampling was conducted in Sokoto, Kebbi and Zamfara states due to significant poultry production in these areas.

## Study design and sample collection

A cross sectional study design was employed to collect 558 pooled shoe socks and dust samples from 165 commercial poultry farms. On arrival to a farm, a pen was randomly selected from other pens as a sampling unit. From this representative unit (pen), sock samples were obtained by stepping on freshly dropped faeces while walking through the pen. Shoe covers were worn over fully covered leather shoes and were changed between farms using clean latex gloves. Shoe socks sample were immediately transferred into a sterile sampling bottle. Additionally, dust samples from that same pen were obtained from multiple spots by scooping up dust materials containing poultry litter materials, feeds and other composed materials into a sterile sample-bottle. The number of samples collected per farm depended on the categories of chicken raised in the farm. Two samples were collected per farm from 51 farms that reared either broilers or layers, while four samples were collected from 114 farms, two from each category of layers and broiler. Information about age of flock, chicken type and category of farm was recorded. A farm was consider positive when at least one of the samples collected was found to contain *Salmonella* species. All samples were adequately labelled and placed in cooling box containing ice-packs. Samples were transported to the laboratory in Central Veterinary Research Laboratory, Usmanu Danfodiyo Univesity, Sokoto, Nigeria for immediate analysis.

## Farm description

Poultry production system could be categorised in to five intermediate categories from the four operational classes of Food and Agricultural Organization (FAO), based on the number of chicken raised in a farm [18]. The size of farms ranged from backyard farms (less than 200 birds), semi- commercial farms (200–999 birds), small-scale farms (1,000–4,999 birds), and medium-scale farms (5,000–9,999 birds) to large-scale farms (more than 10,000 birds). The backyard farms represent the majority of the farms sampled in the study (S1 File). Grand-parent stocks are generally imported from Europe and breeding farms are concentrated outside the study area in the south of Nigeria. Day-old-chicks are likewise mostly produced in the south by big hatcheries and transported by road to different parts of the north-west Nigeria [18].

## Isolation and characterization of *Salmonella*

Samples were investigated for presence of *Salmonella* according to ISO 6579 [27]. Briefly, one gram of sample was weighed (OHAUS, USA) before mixed with 9 ml of buffered peptone water (BPW, Oxoid UK) for non-selective pre-enrichment of samples at 37˚ C for 18 ± 2 hrs. Subsequently, an aliquot of 0.1 ml of the suspension was inoculated into 10 ml of Rappaport-Vassiliadis, (RV) broth (Oxoid, UK) for selective enrichment overnight at 41.5˚C. Then selective plating was done in parallel on Xylose Lysine Deoxycholate, XLD (Oxoid, UK) and onto Brilliance *Salmonella* Agar, BSA (Oxoid, UK); plates were incubated at 37˚C overnight. Plates were examined for the presence of *Salmonella* typical colonies, identified with a black centre or purple colour on XLD and BSA, respectively. One isolate was picked from a pure culture representing one sample unit. The reference strain *Salmonella* Typhimurium ATCC 14028 was spiked into selected samples for quality control purposes.

Presumptive *Salmonella* isolates were subjected to biochemical tests using commercially available media (Oxoid, UK). Briefly, a loopful of colonies was stabbed into citrate and sulphide, indole, motility (SIM) agar, and incubated at 37˚C overnight. Isolates showing positive

citrate, $H_2S$ production, and motility but a negative indole reaction were categorized as presumptive *Salmonella* and sub-cultured onto Nutrient agar (Oxoid, UK) and incubated at 37˚C overnight. Colonies from this plate were subjected to serological confirmation by slide agglutination test using polyvalent *Salmonella* antisera (SSI, Denmark) and normal saline as a negative control and *Salmonella* Typhimurium ATCC 14028 as a positive control.

## PCR-based *Salmonella* identification

As a final confirmation of *Salmonella*, isolates that were positive by serology were subjected to PCR identification using the *invA*-based method [28]. Briefly, one to two bacterial colonies were suspended into 100 µL of molecular grade water (Gibco, Life technologies, USA) and subjected to boiling at 100˚C for 10 min. The mixture was centrifuged (Eppendorf, AG Germany) at 12,000 rpm for 2 min. PCR was performed using PuRe Taq Ready-To-Go PCR beads (illustra TM United Kingdom) containing buffers, dNTPs, enzyme, stabilizers and BSA in addition to 1 µL of sample DNA and 0.2 µL of the primers (inqaba biotec, Hartfield South Africa) (100 µM) *invA* forward (5'GTGAAATTATCGCCACGTTCGGGCA3') and *invA* reverse (5'TCATCGCACCGTCAAAGGAACC3') in 25 µl final volume reaction. Amplification was performed using Thermal cycler (Applied Biosystem, USA) with 95˚C for 2 min, 95˚C for 30 sec, 55˚C for 30 sec and 72˚C for 2 min for 35 cycles. A final cycle at 72˚C for 5 min was used [29]. Amplicons were visualized in 1.5% agarose gels stained with SafeView nucleic acid stain using a UV trans-illuminator (UVP GelMax Imager, United Kingdom). Isolates that showed a band size of 284 bp was considered as *Salmonella* using 100 bp standard DNA ladder (New England Bio-Labs, United Kingdom). The reference strain *Salmonella* ATCC 14028 was used as positive control and water without DNA as negative control.

## Serotype PCR of strains

Initial screening of isolates using serotype specific PCR was done at the Pharmaceutical Microbiology Laboratory University of Ibadan, Nigeria to identify *S.* Enteritidis and *S.* Typhimurium, which are some of the common non- typhoidal *Salmonella* in humans in the region [30].

The protocol developed by Tennant et al. (2010) was used to amplify specific genomic regions of strains to investigate whether they belonged to serotypes *S.* Enteritidis or *S.* Typhimurium; the *SdfF* and *SdfR* primers (inqaba biotec, Hartfield South Africa) were used to amplify SdfI, indicative of *S.* Enteritidis. Two sets of primers, FFLIB and RFLIA (inqaba biotec, Hartfield South Africa), which amplify the *fliB-fliA* intergenic region, and primers Sense-59 and Antisense-83, which amplify the Phase 2 (*fljB*) flagella gene, were used to detect *S.* Typhimurium including the monophasic variant [29, 31, 32]. PCR conditions and procedures were set as described above with primer concentrations of 1 µl each of 0.5 µL *sdf*F/ *sdf*R (5'CGTTCTTCTGGTACGATGAC3' forward, 5'TGTGTTTTATCTGATGCAAGAGG3' reverse), *FFLIB/ RFLIB* (5'GCGGTATACAGTGAATTCAC3' forward, 5'CTGGCGACGATCTGTCGATG3' reverse) sense-59/ Antisense-83 (5'GCCATATTTCAGCCTCTCGCCCG3' forward, 5'CAACAACAACCTGCAGCGTGTGCG3' reverse) for 100 µl reaction final volume respectively. Isolates that showed a band size of 333 bp and 1389/250 were considered as *S.* Enteritidis and *S.* Typhimurium respectively using 100 bp standard DNA ladder (New England BioLabs, United Kingdom).

## DNA extraction and WGS analysis

Single colony of *Salmonella* on blood agar grown over night was suspended in 5 ml Luria broth (LB) (Difco, USA) for 16 hrs at 37˚C in an incubator shaker (GFL, Germany). Genomic DNA was extracted using Promega Maxwell DNA automatic extraction robot and Maxwell

RSC Cultured Cells DNA kit as described by the protocol of the manufacturer (Maxwell® RSC-16, USA). The concentration and quality of extracted DNA was evaluated using Nano-drop (Thermo Scientific, USA), with DNA concentration of greater 20 ng/μL and A260/A280 of 1.8–2.0 were sequenced. A sequencing library was prepared using Nextera XT kits as described by the manufacturer. Genomes were sequenced on an Illumina MiSeq platform using paired-end chemistry (2 x 250-bp) (Illumina, San Diego, California, USA). *De novo* genome assembly of sequence was done using SPAdes version 3.9 available on the Centre of Genomic Epidemiology server (cge.cbs.dtu.dk/services/SPAdes/). The quality of the assembled genome was evaluated using QUAST [33]. The draft genome sequences are available at the European Nucleotide Archive under study accession number PRJEB37477 (secondary accession ERP120792) and accession number for each genome is indicated in S2 File.

### *In Silico* serotype and STs prediction

Because of high-throughput, and decreasing cost of next generation sequencing, WGS based serotyping is increasingly used as methods in *Salmonella* typing [34]. This method has been validated and found to be highly concordance with the results from conventional serotyping methods [17], with better efficiency. Assemblies with a genome size less than 4 Mb or greater than 6 Mb or with GC content of the genome less than 50% or greater than 54% were excluded (S1 Table). Also contaminated and genome assigned to different organism were excluded. Draft assembled genomes of *Salmonella* that satisfied the inclusion criteria were initially uploaded to the online version of SeqSero 2 v1.0.2 ( http://www.denglab.info/SeqSero2) [9, 35]. However, as some strains would not be assigned to serotypes by SeqSero2, draft assemblies were also uploaded to SISTR (https://lfz.corefacility.ca/sistr-app/) through the web application programming interface and the results of the predicted serovars were compared with that of SeqSero2 [11, 16]. Most of the strains were assigned multi-locus sequence types (STs) by SISTR pipeline using seven housekeeping genes (*aro*C, *dna*N, *hem*D, *his*D, *pur*E, *suc*A, *thr*A) [36]. Some isolates could not be assigned ST type by SISTR; raw reads of these strains were submitted to Enterobase (http://enterobase.warwick.ac.uk/).

### Risk factors analysis

A signed written consent was obtain from farmers prior to administration of questionnaire. A questionnaire (S3 File) and consent to collect information about risk factors for *Salmonella* at the poultry farms was designed and pre-tested with a small population of 10 farmers for validity and reliability before applied to 65 consented farmers. The questionnaire contained information about farm manager demography, farm size and management, farmer's knowledge about *Salmonella* and salmonellosis, disease management, farm sanitation and biosecurity (S4 File). The interviews were done during the visits to the farm when the different samples for *Salmonella* analysis were collected. The questions were posed to the owner, farm manager, consulting veterinarian or animal health workers who were available at the time of the visit.

### Data and statistical analysis

Serotype predictions by two pipelines were imported to SPSS version 26 (IBM, USA) to check for level of agreement between the two pipelines using Cohan's kappa statistics. Questionnaire responses were entered into Epi Info 7 (CDC, USA) and later exported to Microsoft Excel 2016 (Microsoft Corporation, Redmond, WA, USA) as a database. Risk factor analysis was done using Statistical software R using (Glm package) relevant installed packages [37]. Chi-square test of independence was used to test for association between *Salmonella* prevalence and categorical variables (farm category, type of chicken, sample and age of chicken). A two-

step statistical procedure was used to evaluate relationship between variables and *Salmonella* farm status. In the first step, 11 potential risk factors (production system, report of previous outbreaks, frequency of Salmonellosis outbreaks, report of Salmonellosis outbreak in neighbouring farm, fencing of farm, poultry waste disposal, proximity with other poultry farms, provision of disinfection of boots, availability of toilets, presence of other livestock in the farm and frequency of farm cleaning) were selected for univariate regression analysis between specific variable and outcome of *Salmonella* status in a farm. In the second step, statistically significant predictors were selected for multiple logistic regression analysis to model between predictors and outcome. The significant level was $p < 0.05$ with results expressed as estimates and standard error.

## Results

### Prevalence of *Salmonella*

Among 165 commercial farms sampled, 47.9% (CI$_{95}$ [40.3–55.5]) were positive for *Salmonella*, while 15.9% (CI$_{95}$ [12.9–18.9]) of the individual samples were positive (Table 1).

Large-scale farms had significantly higher ($p = 0.0001$) *Salmonella* prevalence (33%; CI$_{95}$ [29.1–36.9]) than other farm categories, while small-scale farms had the lowest prevalence. Layer chickens had significantly higher prevalence (20.6%; CI$_{95}$ [17.2–24.0]) than broilers (10.9%; CI$_{95}$ [8.3–13.5]) ($p = 0.003$). Sample type (shoe socks, dust) and age categories were not significantly associated with the prevalence of *Salmonella* in poultry farms (Table 2).

### Serotypes identified in poultry flocks

Characteristics of genomes submitted to *in silico* serotype prediction, and which failed the quality check, are shown in S1 Table. Seventy-four isolates were sequenced, and twenty-three serotypes, all belonging to *S. enterica* subspecies *enterica* were predicted from this analysis. Fourteen isolates could not be assigned serotypes by SeqSero2, but their serotype was predicted by SISTR. One isolate was assigned the same antigenic formula, but both pipelines did not predict the serotype. Seqsero2 uses the new antigenic numeric designation for O antigen, while SISTR use letters for O antigen nomenclature. Multiple serotype predictions were observed for four isolates, while two isolates had double prediction with 66 isolates having a unique serotype assigned. Cohen's Kappa test was run to determine if there was an agreement between SeqSero2 and SISTR serotype predictions. There was a substantial agreement between the two pipelines (k = 0.76, $p < 0.005$). The serotypes and ST types obtained for individual isolates are shown in the S2 Table. Fifteen isolates, whose STs could not be predicted by SISTR, were assigned STs by Enterobase. All strains from same serotype belonged to a single ST. Among the 74 strains, *S.* Kentucky (ST-198) and *S.* Isangi (ST-216) appeared with the highest

**Table 1. Prevalence of *Salmonella* in poultry farms in Nigeria.**

| State | No of farms | No. of samples | [a]*Salmonella*-positive farms | | [b]*Salmonella*-positive samples | |
|---|---|---|---|---|---|---|
| | | | Count | (%) | Count | (%) |
| Sokoto | 62 | 200 | 30 | 48.4 | 33 | 16.5 |
| Kebbi | 48 | 176 | 17 | 35.4 | 19 | 10.8 |
| Zamfara | 55 | 182 | 32 | 58.2 | 37 | 20.3 |
| **Total** | **165** | **558** | **79** | **47.9** | **89** | **15.9** |

[a]Farm Confidence Interval = CI$_{95}$ (40.3–55.5)

[b]Sample level Confidence Interval = CI$_{95}$ (12.9–18.9)

**Table 2. Variation in prevalence of *Salmonella* based on selected parameters in commercial poultry farms in Nigeria.**

| Parameters | Number sampled | *Salmonella*-positive | | |
|---|---|---|---|---|
| **Farm categories** | | Count | % | *p*-value |
| Backyard | 119 | 18 | 15.1 | *p* = 0.0001 |
| Semi-commercial | 81 | 18 | 22.2 | |
| Small-scale | 198 | 11 | 5.6 | |
| Medium-scale | 66 | 11 | 16.7 | |
| Large-scale | 94 | 31 | 33.0 | |
| **Sample type** | | | | |
| Shoe socks | 279 | 43 | 15.4 | *p* = 0.82 |
| Dust | 279 | 46 | 16.5 | |
| **Chicken type** | | | | |
| Layers | 292 | 60 | 20.6 | *p* = 0.003 |
| Broilers | 266 | 29 | 10.9 | |
| **Age category** | | | | |
| Broiler Starter | 90 | 11 | 12.2 | *p* = 0.78 |
| Broiler Finisher | 176 | 18 | 10.2 | |
| Chicks | 28 | 6 | 21.4 | *p* = 0.19 |
| Growers | 50 | 5 | 10.0 | |
| Layers | 212 | 49 | 23.1 | |
| Spent layers | 2 | 0 | 0.0 | |

prevalence, 32.8% and 11% respectively, while *S.* Poona (ST-308), *S.* Virchow (ST-6166) and *S.* Waycross (ST-7745) were among the serotypes with lowest frequencies (1.4%) observed (Table 3).

Serotyping remains the first step to characterize *Salmonella* isolates [5]. However, the traditional phenotypic method for serotyping is logistically challenging, as it requires the use of more than 150 specific antisera and well-trained personnel to interpret the results, and it may show low performance due to weak or non-specific agglutination, auto-agglutination or loss of antigen expression [38], which may lead to delay in rapid identification and false prediction of serovars involved in an outbreak. Alternative methods based on PCR amplification of specific genomic regions of O and H antigens were developed [39]. In view of this, we evaluated PCR based method [29] for serotyping. The result of serotype-specific PCR for *S.* Enteritidis showed that 13/73 isolates were *S.* Enteritidis, however, these were assigned different serotypes by WGS (four assigned to *S.* Kentucky, two to *S.* Chester and seven to other different serotypes). No strain was found positive in the *S.* Typhimurium-specific PCR (S2 Table).

Spatial variation in the distribution of serotypes was evident. *S.* Larochelle (ST-22), *S.* Abadina and *S.* Telekebir (ST-2222) were exclusively identified in Zamfara state, while *S.* Schwarzengrund (ST-96) and *S.* Muenster (ST-321) were only identified in Sokoto state. Likewise, *S.* Takoradi (ST-531) and *S.* Poona (ST-308) were only identified in Kebbi state. However, *S.* Kentucky appeared with the highest prevalence in all three states and *S.* Isangi was common in Sokoto and Zamfara states (Fig 1).

## Risk factors for presence of *Salmonella* in poultry farms

In the first univariate analysis of covariates from farm data, five factors were significantly associated with prevalence of *Salmonella* at the farm ($p < 0.05$); i.e. production system, report of salmonellosis outbreak in neighbouring farm, on-farm disposal of poultry waste, proximity to other poultry farms and presence of other livestock at the farm. In contrast, fencing of farm,

**Table 3. Frequency distribution of *Salmonella* serotypes identified at Nigerian poultry farms.**

| S/N | Serotypes | Number of strains (n = 74) | Percentage (%) |
|---|---|---|---|
| 1 | S. Abadina | 2 | 2.7 |
| 2 | S. Aberdeen | 1 | 1.4 |
| 3 | S. Alachua | 1 | 1.4 |
| 4 | S. Birmingham | 1 | 1.4 |
| 5 | S. Bradford | 1 | 1.4 |
| 6 | S. Chester | 2 | 2.7 |
| 7 | S. Chomedey | 1 | 1.4 |
| 8 | S. Colindale | 1 | 1.4 |
| 9 | S. Corvalis | 2 | 2.7 |
| 10 | S. Esen | 1 | 1.4 |
| 11 | S. Give | 1 | 1.4 |
| 12 | S. Isangi | 8 | 10.8 |
| 13 | S. Ituri | 2 | 2.7 |
| 14 | S. Kentucky | 24 | 32.4 |
| 15 | S. Larochelle | 4 | 5.4 |
| 16 | S. Menston | 1 | 1.4 |
| 17 | S. Muenster | 4 | 5.4 |
| 18 | S. Poona | 1 | 1.5 |
| 19 | S. Schwarzengrund | 4 | 5.4 |
| 20 | S. Takoradi | 6 | 8.1 |
| 21 | S. Telelkebir | 3 | 4.1 |
| 22 | S. Virchow | 1 | 1.4 |
| 23 | S. Waycross | 1 | 1.4 |
| 24 | -:z13,z28:I,z13,z28 | 1 | 1.4 |

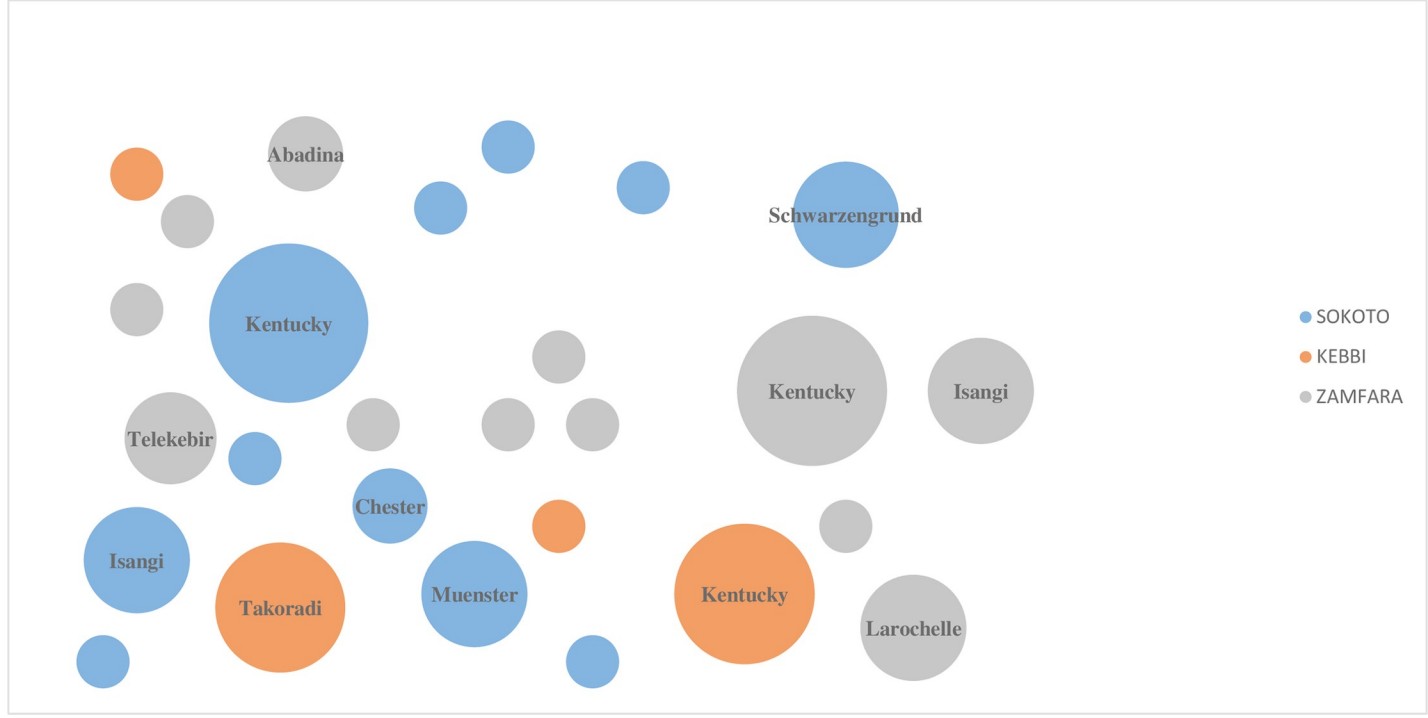

**Fig 1. Spatial bubble graph description of variation of *Salmonella* serotypes identified from poultry farms in different regions of Nigeria (colour marked).** The relative size of the bubble indicates the relative number of strains reported in that particular serovar.

**Table 4. Univariate analysis of variables associated with *Salmonella* infection in poultry farms in Nigeria.**

| Variables | Responses (n = 65) | Positive for *Salmonella* (%) | Estimate ± SE | *p*-value |
|---|---|---|---|---|
| **Production system** | | | | |
| Deep litter | 19 | 29.2 | 1.89±0.56 | 0.000713 |
| Battery cage | 8 | 12.3 | | |
| **Previous outbreaks** | | | | |
| Yes | 27 | 41.5 | 20.46±2069.61 | 0.992 |
| No | 0 | 0 | | |
| **Frequency of outbreak** | | | | |
| None | 0 | 0 | -20.95± 2109.0 | 0.9921 |
| Once | 3 | 4.6 | -1.90±0.92 | 0.0391 |
| Twice | 8 | 12.3 | -0.41±0.89 | 0.6442 |
| More | 16 | 24.6 | | |
| **Outbreaks at neighbouring farms** | | | | |
| Yes | 17 | 26.2 | 2.99± 0.72 | 3.48e-05 |
| No | 10 | 22.2 | | |
| **Farm fenced** | | | | |
| Yes | 6 | 9.2 | -3.40±0.70 | 1.37e-06 |
| No | 21 | 32.3 | | |
| **Waste management** | | | | |
| On farm | 24 | 36.9 | 3.75±0.76 | 7.09e-07 |
| Off farm | 3 | 4.6 | | |
| **Presence of other livestock** | | | | |
| Yes | 24 | 36.9 | 3.40±0.73 | 3.2e-06 |
| No | 3 | 4.6 | | |
| **Proximity with farms** (~1 km) | | | | |
| Yes | 20 | 30.8 | 2.08±0.57 | 0.000286 |
| No | 7 | 10.8 | | |
| **Disinfection of boots** | | | | |
| Yes | 3 | 4.6 | -3.97±0.78 | 3.43e-07 |
| No | 24 | 36.9 | | |
| **Lavatory** | | | | |
| Yes | 2 | 3.1 | -3.85±0.84 | 4.14e-06 |
| No | 25 | 38.5 | | |
| **Cleaning frequency** | | | | |
| Weekly | 1 | 1.5 | -3.72±1.08 | 0.000603 |
| Yearly | 7 | 10.8 | 18.11± 2465.3 | 0.994140 |
| Monthly | 19 | 29.2 | | |

provision of boot disinfection and staff lavatory were negatively associated with the prevalence of *Salmonella* (Table 4).

In the second step logistic regression analysis, on-farm waste disposal and presence of other livestock in a farm showed statistically significant association with *Salmonella* infection. Using the logit model, the positive coefficient of the estimates indicated that disposing poultry waste on farm was associated with a three-fold higher chance that the farm was positive for *Salmonella*, while presence of other livestock increased the log odds by 2.6 units (Table 5).

**Table 5. Logistic regression model of risk factors for presence of *Salmonella* in farms in Nigeria.**

| Predictors | Estimate | ± SE | *p* value |
|---|---|---|---|
| Intercept | -5.0811 | 1.4741 | 0.000567 |
| Production system | | | |
| Battery cage | 1.4472 | 1.1465 | 0.206834 |
| Neighbouring outbreak | | | |
| Yes | 1.6299 | 1.2170 | 0.180491 |
| Waste management | | | |
| On farm | 3.2436 | 1.1710 | 0.005605 |
| Presence of other livestock | | | |
| Yes | 2.6157 | 1.1001 | 0.017425 |
| Proximity with farms (~1 km) | | | |
| Yes | 0.7638 | 1.1249 | 0.497120 |

## Discussion

In this study, a high farm prevalence (47.9%) of *Salmonella* infection was observed in commercial poultry farms in Nigeria. The results confirms observations from other parts of Nigeria by [21] who showed 43.6% farm prevalence in commercial layer farms. Relative high farm prevalence have also been reported in other sub-Saharan countries such as Ghana (44.0%), Uganda (20.7%), and Ethiopia (14.6%) [40–42] and likewise in developing Asian countries with report of 46.3% and 18% prevalence in central Vietnam and Bangladesh respectively [3, 43]. This is in contrast to many developed countries like Poland, where the total percentage of infected flocks was 1.57%, and where a decrease in prevalence of *Salmonella* spp. in broiler chickens was observed from 2.19% in 2014 to 1.22% in 2016. In Denmark, the prevalence for *Salmonella* infection poultry has been very low (0% to 1.8%) in the last decade, with the highest flock prevalence of 2.6% recorded in 2018 [44]. The reduction in European member countries can be attributed to implementation of specific control programmes [45], which are lacking in developing countries like Nigeria. Sample level prevalence (15.9%) of *Salmonella* from this study was similar to previously reported prevalence in other parts of Nigeria by Fagbamila et al., (2017) but higher than reported by Eguale (2018) (14.1% and 4.7% sample prevalence, respectively). Large scale farms were found to have higher *Salmonella* sample prevalence compared to other categories of farm levels, indication that once large farms were infected, the infection became more widespread in this farm type. Adesiyun et al. [46] observed a similar tendency for large farms compared to other farm categories from Caribbean countries. This might be attributed to large number of the flock making it difficult for the farmer to adhere to strict farm bio-securities and good farm management practices. The observations is not surprising, since there is conclusive evidence by European Food Safety Authority that larger poultry farms have higher chances of increased occurrence, persistence and spread of *Salmonella* [47, 48]. Furthermore, layer flocks, which spend longer time in the poultry house, had higher prevalence of *Salmonella* infection compared with broiler flocks. Wierup et al. [49] have also showed a substantially higher prevalence of *Salmonella* in layer flocks than in broilers among outdoor and indoor housing system.

A high number of *Salmonella* serotypes were observed in the farms investigated suggesting either a wide diversity of sources for introduction of *Salmonella* into the farms, or that common sources (such as contaminated feed) can contain different serotypes over time. Reports from other countries have also showed diversity in serotypes of non- typhoidal *Salmonella* in poultry farms [41, 42, 50–52]. Notably, *S*. Enteritidis was absent. This may be because the

available vaccine used against *Salmonella* Gallinarum confers cross protection against other group D-strains [21, 53]. Also *S*. Typhimurium, which is commonly associated with poultry [54], was not observed in this study. This confirms observations by Fagbamila et al., (2017) and Useh et al., (2016) that these two serotypes play marginal role in the poultry industry in Nigeria.

*S*. Kentucky was the most commonly observed serotype. This serotype apparently has poultry as the main reservoir [55, 56], was also isolated from an health cattle [57]. And it has, over the years, emerged as a global zoonotic pathogen [58]. Human illnesses caused by this pathogen in North America and Europe are typically associated with a history of travel to Africa, Southeast Asia, and the Middle East, where this pathogen is established in poultry [59].

In general, the study observed predominantly *Salmonella* C group of the WKL scheme with few other members of B, E, G, O and S groups. Commonly isolated serotypes, besides *S*. Kentucky (ST-198), included *S*. Schwarzengrund (ST-96), *S*. Muenster, *S*. Poona, *S*. Isangi, *S*. Chester and *S*. Virchow. *S*. Schwarzengrund has been recorded from human in Denmark and the United States, where several isolates have shown multidrug resistance [54]. Recently it was isolated from diarrheal patients in a food poisoning event in China [60]. It was the fifth most common serovar isolated from retail meat in the United States in 2004, associated exclusively with poultry products, and other studies also suggest that poultry could be the most common reservoir [54]. *S*. Muenster has mainly been associated with salmonellosis in cattle [61]. In the current study, presence of other livestock in a farm was identified as a risk factor for *Salmonella* occurrence in poultry, and it may be that isolation in poultry is associated with horizontal transmission from other livestock. This serotype was associated with a nationwide outbreak of gastrointestinal illness in France, 2008 [62]. *S*. Poona has been reported from multistate outbreak in United States in 2015–2016 and was linked with the consumption of cucumber [55]. Reports from poultry are not common. Similarly, *S*. Isangi was isolated from a nosocomial infection outbreaks [63], while *S*. Chester accounted for 0.1% of all annual human salmonellosis cases notified in the EU/EEA [64]. *S*. Chester was also the second most common serotype in poultry, in 2010, in Burkina Faso [65]. *S*. Virchow is a serotype associated with poultry [66] and was reported to cause typhoid-like illness with fever and altered consciousness in human blood and stool culture [67]. This study also observed spatial variation in the distribution of serotypes, with some serotypes dominating a particular geographical area. This may reflect an ecologic niche established by those serotypes restricting them to a particular geographical region [68]. Similarly reported by Li et al. [69] and Pointon et al. [70] observed the dominance of one serovar over others in a particular geographical area.

Since Public Health England implemented whole genome sequencing (WGS) as a routine typing tool for public health surveillance of *Salmonella* [7], the use of WGS data for *Salmonella* serotyping has increased steadily. The method depends on publicly available databases. There was a substantial agreement between SeqSero2 and SISTR predictions (k = 0.76), and all the serotypes predicted by SeqSero2 were adequately predicted by SISTR. However, 14 isolates could not be predicted by SeqSero2 due to problems with adequate identification of O antigens. Six of these isolates were assigned multiple serovars in SISTR, while the serotype of the remaining isolates was resolved by this prediction too. Diep et al. (2019) likewise observed (1%) incomplete predictions of serotype when SeqSero2 was used, and explained this to be due to the same antigenic formula shared by strains from different subspecies, and that some serotypes in the WKL scheme require additional phenotypes for differentiation. Additionally, some serotypes in the WKL scheme differ only by minor epitopes of the same O antigen group. SISTR, in addition to using somatic (O) and flagella (H), utilizes the 330 genes in the SISTR cgMLST scheme, which provide an approximation of the genetic distance between serovars. This approximation is useful for disambiguating serovars with similar antigenic formula

[16]. A recent study which assessed the performance of *in silico* serotyping of *Salmonella* spp. found the best performing prediction tool to be SISTR with 94% accuracy, followed by Seq-Sero2 (87%) [71]. However, SISTR could not assign ST types to some isolates, and these were assigned by Enterobase platform. This could be attributed to the fact that, Enterobase MLST database is synchronized and updated daily from pubMLST and other public databases [72], making the platform a more robust portal to get STs for large number of isolates.

There was discordant between the serotypes assigned by both *in silico* pipelines and the result from PCR serotyping. This may be due to the fact that, the primers (*Salmonella* difference fragment, *Sdf* I) we used to amplify our strains could also anneal to other genomic region in other serotypes. This could be explained by the report of Tennant et al. (2010) at the initial validation of the primers that, observed faint band amplicon products of the size of *Sdf* I in *S*. Meleagridis and *S* Livingstone. The *Sdf* genes was reported to be absent in only 34 serovars of *Salmonella* [32], so there is every possibility that some of these serotypes are among the remaining *Salmonella* serovars that possess the *Sdf* gene. The PCR-based method may be particularly unsuitable for assessing serotypes of livestock, wild-life and environmental isolates, as diverse serovars are often prevalent in these niches.

The finding from this study showed that *Salmonella* occurrence in poultry farms was influenced by several risk factors. In the final multivariate modeling, practicing deep liter system was observed as an important risk factor for the prevalence of *Salmonella* infection. The possible explanation could be that farmers seldom clean their deep litter poultry pen, which may lead to the persistent of *Salmonella* in poultry litter. Survival and persistence of *Salmonella* has been observed for 18 months in poultry litter [73, 74] which might result in higher chances of *Salmonella* infection than in battery cage system. A Study conducted by Mollenhorst et al. [75] showed that farms on deep litter system has a significant increased risk of *Salmonella* infection. Furthermore, farms located with close proximity with other farms and with report of neighboring farms having outbreaks of poultry salmonellosis was observed to have significantly increased risk of *Salmonella* infection. This could be due to personnel interaction and sharing of farm equipment, which could possibly introduce bacteria through contaminated tools or persons as previously described by Namata et al. (2009). Airborne transmission could also account for this risk factor, even though, based on available literature, aerosols do not appear to be important in the spread of salmonellosis. It has been earlier speculated that reported that *Salmonella* could become airborne, remain viable in the air and get transmitted among livestock over short distances [76, 77].

Improper waste disposal was observed to be at higher risk for infection with *Salmonella* in poultry farm as it allow for possible re-introduction of *Salmonella* through fomites into the poultry pen after cleaning. Furthermore, presence of other livestock in the farm was also identified as a risk factor to *Salmonella* prevalence. This could simply be explained by detection of serotypes that were associated with other farm animals like *S*. Muenster; a serotype which is associated with cattle [78]. This particular finding completely agrees with the study conducted by Djeffal et al. [79] who observed the presence of other livestock in a poultry farm as a risk factor to *Salmonella* infection.

## Conclusion

Taken together, a high prevalence of *Salmonella* was observed in commercial poultry farms in Nigeria. Importantly, based on WGS data obtained in this study, we showed that a diverse non-typhoidal *Salmonella* serotypes circulate in commercial poultry farms in the study area with *S*. Kentucky (ST-198) having the highest prevalence and the widest geographical coverage. We also showed that WGS based serotyping with SISTR platform had higher chance of

assigning serotypes than SeqSero2. Finally, presence of other livestock on farms and improper poultry waste-disposal have been identified as factors that increases the risk of having *Salmonella* infection in a farm.

## Supporting information

**S1 Table. Quality assurance of 89 genomes assemblies for inclusion into the study.**
(DOCX)

**S2 Table. Genomic characteristics and serotype predictions of *Salmonella* strains isolated from poultry in Nigeria.**
(DOCX)

**S1 File. Sample collection metadata.**
(XLSX)

**S2 File. Sequence assembled genome characteristics and strain accession number.**
(XLSX)

**S3 File. Questionnaire template to assess risk factor of *Salmonella* infection in commercial poultry farms in Northwest, Nigeria.**
(DOCX)

**S4 File. Farmer's questionnaire responses.**
(XLSX)

## Author Contributions

**Conceptualization:** Iruka N. Okeke, Anders Dalsgaard, John Elmerdahl Olsen.

**Data curation:** Egle Kudirkiene.

**Funding acquisition:** Iruka N. Okeke, Anders Dalsgaard, John Elmerdahl Olsen.

**Investigation:** Abdurrahman Hassan Jibril, Olabisi Comfort Akinlabi, Muhammad Bashir Bello.

**Methodology:** Abdurrahman Hassan Jibril.

**Supervision:** Iruka N. Okeke, Anders Dalsgaard, John Elmerdahl Olsen.

**Writing – original draft:** Abdurrahman Hassan Jibril.

**Writing – review & editing:** Anders Dalsgaard, John Elmerdahl Olsen.

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
