## [Decision Letter · Decision Letter 0]

15 Jun 2020

PONE-D-20-10479

Prevalence and risk factors of Salmonella in commercial poultry farms in Nigeria

PLOS ONE

Dear Dr. Olsen,

Thank you for submitting your manuscript to PLOS ONE. After careful consideration, we feel that it has merit but does not fully meet PLOS ONE’s publication criteria as it currently stands. Therefore, we invite you to submit a revised version of the manuscript that addresses the points raised during the review process.

The manuscript has been reviewed by two reviewers and both found interest in it. There are some issues to solve though as the materials and methods should be more complete. Also, further analysis on the WGS data should be performed. It should include more analysis, as the antimicrobial resistance, virulence, plasmids,... make a complete analysis of the WGS data you have.

We look forward to receiving your revised manuscript.

Kind regards,

Patrick Butaye, DVM, PhD

Academic Editor

PLOS ONE

Journal Requirements:

2. In your Methods section, please provide additional location information of the study area, including geographic coordinates for the data set if available.

3. We note that you are reporting an analysis of a microarray, next-generation sequencing, or deep sequencing data set. PLOS requires that authors comply with field-specific standards for preparation, recording, and deposition of data in repositories appropriate to their field. Please upload these data to a stable, public repository (such as ArrayExpress, Gene Expression Omnibus (GEO), DNA Data Bank of Japan (DDBJ), NCBI GenBank, NCBI Sequence Read Archive, or EMBL Nucleotide Sequence Database (ENA)). In your revised cover letter, please provide the relevant accession numbers that may be used to access these data. For a full list of recommended repositories, see http://journals.plos.org/plosone/s/data-availability#loc-omics or http://journals.plos.org/plosone/s/data-availability#loc-sequencing

4. Please provide additional details regarding participant consent.

In the ethics statement in the Methods and online submission information, please ensure that you have specified (i) whether consent was informed and (ii) what type you obtained (for instance, written or verbal, and if verbal, how it was documented and witnessed).

Reviewers' comments:

Reviewer's Responses to Questions

**Comments to the Author**

1. Is the manuscript technically sound, and do the data support the conclusions?

Reviewer #1: Yes

Reviewer #2: Yes

2. Has the statistical analysis been performed appropriately and rigorously? 

Reviewer #1: Yes

Reviewer #2: Yes

3. Have the authors made all data underlying the findings in their manuscript fully available?

Reviewer #1: Yes

Reviewer #2: No

4. Is the manuscript presented in an intelligible fashion and written in standard English?

Reviewer #1: Yes

Reviewer #2: Yes

5. Review Comments to the Author

Reviewer #1: This study was designed to quantify the prevalence of Salmonella in poultry production in Nigeria. In order to understand the Salmonella serotypes most often associated with poultry production, whole genome sequencing was used to serotype isolates collected from shoe sock and dust samples from 165 poultry farms in northwest Nigeria. The data show that Salmonella is commonly found on poultry farms in Nigeria and that Salmonella Kentucky is ubiquitous. In general, this paper is well-written and represents important data that is timely and relevant.

General Comments

1. Some minor typos and grammatical issues throughout. Suggest a thorough edit during revisions.

2. The methods are lacking sufficient detail for the study to be repeatable. Specific comments are included below.

3. Why was serotype PCR and WGS done for serotyping?

4. The terminology used in this manuscript needs attention, specifically as it refers to strains. More details below.

Specific Comments

Study design and sample collection

1. if a farm raised broilers OR layers, then 2 samples were collected per farm. However, if a farm raised BOTH, then 4 samples were collected per type, for a total of 8 samples on the farm? If this is true, then why were double the number of samples collected on farms that had both types of birds? If this is not true, then suggest rewording this to be less confusing. Also, is this only referring to the number of dust samples collected? Or does it also include shoe socks? In general, the sampling scheme is not clear. How did the authors end up with 558 total samples?

2. Species should not be italicized.

3. What are composed materials?

4. How were the dust samples collected? From one spot in the pen? Multiple spots in the pen? Was there a strategy to ensure consistency and representative sampling?

5. How were the shoe sock samples handled? Placed into a sterile sample bag?

Farm Description

1. It seems like the sampling strategy should have been developed to also consider farm size. More samples taken from larger farms. Was this considered?

Isolation and Characterization of Salmonella

1. How was the shoe sock sampled? The methods state 1 gram was added to BPW, but how was 1 gram obtained from the shoe sock?

Risk Factor Analysis

1. Why were only 65 farmers surveyed when a total of 165 farms were sampled?

Results

1. Several instances where Salmonella is not italicized in the tables.

2. Above table 2, sample types are referred to as (faeces, dust). Why is faeces now being mentioned when shoe socks have been the focus and are what is mentioned in the table?

3. Age has not been mentioned previously as a variable of interest. What was the sampling strategy for this? Or, how was this included in the study?

4. In reference to serotyping, the authors are using the term "strains". Shouldn't this be isolates? For example: "Seventy-four strains were sequenced, and a total of 23 serotypes..." In this instance, shouldn't it be that 74 isolates were sequenced?

5. How were isolates pulled from the plates? Were isolated colonies picked and re-streaked to ensure purity? At one point, it is stated the "Multiple serotype predictions were observed for four strains..." It is assumed that the authors mean that multiple serotypes were predicted for four different isolates. Is this true? If so, this goes back to the initial question asking about re-streaking and purity. Could it be possible that the isolates were not pure and could have represented multiple serotypes?

6. It's not clear if 1 isolate was pulled per sample or if multiple. These types of details are lacking and should be included.

7. When referring to multiple isolates within a single serotype, it would be more appropriate to refer to these as strains; however, it is difficult to know if different strains exist within a single serotype without further characterization, such as WGS.

8. It is difficult to interpret the results section with the term strain being used to mean isolate.

Discussion

1. Salmonella Kentucky has also been found in cattle.

2. Once again, the authors need to pay particular attention to the use of isolate, strain, and serotype. For example, on page 20, the authors state that "S. Virchow is a strain associated with poultry..." However, the term serotype would be most appropriate here.

Reviewer #2: Hello!

I liked this. I think you can do additional analyses with the WGS and look at things like antibiotic resistance and virulence as well, but that is up to you (it would be a stronger paper). The written language is a bit rough and could use a good revision as tense and subject/verb disagreement is common place.

Usually, data like this comes out from various parts of the world and it is not complete, this is good.

A quick question though, socks? Do you mean boot covers or did someone actually step on socks? Were their feet washed between barns? Why not fresh dropping collections and how did you control for human to sock contamination?

Flush out your methods a bit. Tell me more about the sequencing. If you have a media or reagent, include the company name and city (with state or country) of origin. Be sure to spell out any acronym.

6. PLOS authors have the option to publish the peer review history of their article (what does this mean?). If published, this will include your full peer review and any attached files.

Reviewer #1: No

Reviewer #2: No

---

## [Author Response · Author response to Decision Letter 0]

4 Aug 2020

Journal requirements

Template adhered to

2. In your Methods section, please provide additional location information of the study area, including geographic coordinates for the data set if available.

 Author response: Image of study area provided

3. We note that you are reporting an analysis of a microarray, next-generation sequencing, or deep sequencing data set. PLOS requires that authors comply with field-specific standards for preparation, recording, and deposition of data in repositories appropriate to their field. Please upload these data to a stable, public repository (such as ArrayExpress, Gene Expression Omnibus (GEO), DNA Data Bank of Japan (DDBJ), NCBI GenBank, NCBI Sequence Read Archive, or EMBL Nucleotide Sequence Database (ENA)). In your revised cover letter, please provide the relevant accession numbers that may be used to access these data. For a full list of recommended repositories, see http://journals.plos.org/plosone/s/data-availability#loc-omics or http://journals.plos.org/plosone/s/data-availability#loc-sequencing

 Author response: Accession numbers provided

4. Please provide additional details regarding participant consent.

In the ethics statement in the Methods and online submission information, please ensure that you have specified (i) whether consent was informed and (ii) what type you obtained (for instance, written or verbal, and if verbal, how it was documented and witnessed).

 Author response: Template of written consent provided. Information that only data from farms where farmers gave consent were included in the study is now included in the manuscript.

Reviewer #1: This study was designed to quantify the prevalence of Salmonella in poultry production in Nigeria. In order to understand the Salmonella serotypes most often associated with poultry production, whole genome sequencing was used to serotype isolates collected from shoe sock and dust samples from 165 poultry farms in northwest Nigeria. The data show that Salmonella is commonly found on poultry farms in Nigeria and that Salmonella Kentucky is ubiquitous. In general, this paper is well-written and represents important data that is timely and relevant.

General Comments

1. Some minor typos and grammatical issues throughout. Suggest a thorough edit during revisions.

Author’s response: Thorough edition done.

2. The methods are lacking sufficient detail for the study to be repeatable. Specific comments are included below.

Author’s response: We have addressed the comments under specific comments below.

3. Why was serotype PCR and WGS done for serotyping?

Author’s response: Reasons for using PCR and WGS are now explained in the manuscript. 

4. The terminology used in this manuscript needs attention, specifically as it refers to strains. More details below.

Author’s response: We have generally now use the term isolate.

Specific Comments

Study design and sample collection

1. if a farm raised broilers OR layers, then 2 samples were collected per farm. However, if a farm raised BOTH, then 4 samples were collected per type, for a total of 8 samples on the farm? If this is true, then why were double the number of samples collected on farms that had both types of birds? If this is not true, then suggest rewording this to be less confusing. Also, is this only referring to the number of dust samples collected? Or does it also include shoe socks? In general, the sampling scheme is not clear. How did the authors end up with 558 total samples?

Author’s response: Sampling scheme is now appropriately described as suggested in the manuscript. "Two samples was collected per farm from 51 farms that reared either broilers or layers, while four samples was collected from 114 farms, 2 from each category of layers and broiler".

2. Species should not be italicized.

Author’s response: Corrected in the manuscript

3. What are composed materials?

Author’s response: "Assorted compound composed of bedding materials, animal waste, dead skin, feed scraps, water, and feathers". Sentence modified in the manuscript

4. How were the dust samples collected? From one spot in the pen? Multiple spots in the pen? Was there a strategy to ensure consistency and representative sampling?

Author’s response: Dust samples were collected from multiple spots in the pen to have a representative sampling. Sentence modified in the manuscript

5. How were the shoe sock samples handled? Placed into a sterile sample bag?

Author’s response: Immediately transferred into a sterile sampling bottle. Sentence modified in the manuscript

Farm Description

1. It seems like the sampling strategy should have been developed to also consider farm size. More samples taken from larger farms. Was this considered?

Author’s response: No, because the goal was to estimate Salmonella prevalence for farm and sample. We took a farm as a sampling unit. 

Isolation and Characterization of Salmonella

1. How was the shoe sock sampled? The methods state 1 gram was added to BPW, but how was 1 gram obtained from the shoe sock?

Author’s response: It was weighed, using a weighing scale. Sentence modified in the manuscript

Risk Factor Analysis

1. Why were only 65 farmers surveyed when a total of 165 farms were sampled?

Author’s response: It was based on the number of farmers who provided written consent. This information has been added to the manuscript. Template of written consent is now attached in S3_File

Results

1. Several instances where Salmonella is not italicized in the tables.

Author’s response: Italicised and effected in the manuscript

2. Above table 2, sample types are referred to as (faeces, dust). Why is faeces now being mentioned when shoe socks have been the focus and are what is mentioned in the table?

Author’s response: Corrected

3. Age has not been mentioned previously as a variable of interest. What was the sampling strategy for this? Or, how was this included in the study?

Author’s response: Age of birds in flock was recorded during sample collection. The information on data collection about age is reported in the methods "Study design and sample collection"

4. In reference to serotyping, the authors are using the term "strains". Shouldn't this be isolates? For example: "Seventy-four strains were sequenced, and a total of 23 serotypes..." In this instance, shouldn't it be that 74 isolates were sequenced?

Author’s response: Corrected appropriately in the manuscript

5. How were isolates pulled from the plates? Were isolated colonies picked and re-streaked to ensure purity? At one point, it is stated the "Multiple serotype predictions were observed for four strains..." It is assumed that the authors mean that multiple serotypes were predicted for four different isolates. Is this true? If so, this goes back to the initial question asking about re-streaking and purity. Could it be possible that the isolates were not pure and could have represented multiple serotypes?

Author’s response: No, only single colony from a plate was re-streaked on the media. The pipeline assigned multiple predictions to a single isolate. This is due to sequence similarity between predicted serotypes for an isolate.

6. It's not clear if 1 isolate was pulled per sample or if multiple. These types of details are lacking and should be included.

Author’s response: One isolates per sample. Information added in the manuscript in methods "Isolation and characterization of Salmonella" 

7. When referring to multiple isolates within a single serotype, it would be more appropriate to refer to these as strains; however, it is difficult to know if different strains exist within a single serotype without further characterization, such as WGS.

Author’s response: Okay, we have changed appropriately

8. It is difficult to interpret the results section with the term strain being used to mean isolate.

Author’s response: All corrected

Discussion

1. Salmonella Kentucky has also been found in cattle.

Author’s response: Information updated

2. Once again, the authors need to pay particular attention to the use of isolate, strain, and serotype. For example, on page 20, the authors state that "S. Virchow is a strain associated with poultry..." 

However, the term serotype would be most appropriate here.

Author’s response: Corrected

Reviewer #2: Hello!

I liked this. I think you can do additional analyses with the WGS and look at things like antibiotic resistance and virulence as well, but that is up to you (it would be a stronger paper). The written language is a bit rough and could use a good revision as tense and subject/verb disagreement is common place.

Author’s response: Thank you. We have chosen to separate AMR from Salmonella prevalence. 

Usually, data like this comes out from various parts of the world and it is not complete, this is good.

A quick question though, socks? Do you mean boot covers or did someone actually step on socks? Were their feet washed between barns? Why not fresh dropping collections and how did you control for human to sock contamination?

Author’s response: You put a sterile shoe socks cover over your boots and walk around the pen. It is a standard method of collecting a pool sample from fresh droppings. At each poultry houses we disinfected boots in a foot dip (a concrete made depression of about 10 cm depth and 1.5 meters wide containing water and disinfectant) at the entrance of the pen, before putting the shoe sock cover over the boots. Generally, we maintained and adhered to strict biosecurity measures during sample collection.

Flush out your methods a bit. Tell me more about the sequencing. If you have a media or reagent, include the company name and city (with state or country) of origin. Be sure to spell out any acronym.

Author’s response: Sources of all reagents used and media are adequately indicated. All acronyms are well spelt out

---

## [Editor Report · Decision Letter 1]

12 Aug 2020

Prevalence and risk factors of Salmonella in commercial poultry farms in Nigeria

PONE-D-20-10479R1

Dear Dr. Olsen,

We’re pleased to inform you that your manuscript has been judged scientifically suitable for publication and will be formally accepted for publication once it meets all outstanding technical requirements.

Kind regards,

Patrick Butaye, DVM, PhD

Academic Editor

PLOS ONE
---

## [Editor Report · Acceptance letter]

14 Sep 2020

PONE-D-20-10479R1 

Prevalence and risk factors of *Salmonella* in commercial poultry farms in Nigeria 

Dear Dr. Olsen:

I'm pleased to inform you that your manuscript has been deemed suitable for publication in PLOS ONE. Congratulations! Your manuscript is now with our production department. 

Kind regards, 

on behalf of

Professor Patrick Butaye 

Academic Editor

PLOS ONE